# Static Thermal Model of a Fibre-Optic Rotational Seismometer

**DOI:** 10.3390/s25103184

**Published:** 2025-05-19

**Authors:** Piotr Zając, Marcin Janicki, Cezary Maj

**Affiliations:** Department of Microelectronics and Computer Science, Lodz University of Technology,93-005 Lodz, Poland; piotr.zajac@p.lodz.pl (P.Z.); cezary.maj.1@p.lodz.pl (C.M.)

**Keywords:** opto-electronic sensor, static compact thermal model, fibre-optic rotational seismometer

## Abstract

Fibre-optic rotational seismometers are devices capable of high-accuracy measurements of rotation rates. Taking into account that temperature significantly influences their operation, particularly the optical signal path, accurate thermal models of these devices are required. This paper presents a static thermal model of such a fibre-optic rotational seismometer. This compact thermal model is based on a thermal resistor network, where the temperature values are computed only in the most important locations of the device. The values of thermal model elements are estimated based on thermocouple temperature measurements. Owing to the accurate determination of temperature values, especially in fibre-optic loops, it was possible to achieve the desired device sensitivity.

## 1. Introduction

Fibre-optic rotational seismometers (FORSs) are capable of extremely high-accuracy measurements of rotation rates. These devices are superior to their mechanical counterparts, and their sensitivity can be better than 100 nrad/s, i.e., three orders of magnitude below the Earth’s rotation speed, which is equal to 73 μrad/s. Thus, they can be used for monitoring vibrations induced by earthquakes or explosions [1,2,3,4]. The operating principle of FORSs is based on the Sagnac effect [5,6]. The original mirror-based device was a ring interferometer, in which the light emitted by a collimated source was split into two beams traveling in opposite directions along the same polygonal path producing the interference fringe pattern due to the phase difference Δ*Φ_R_* between the beams. Currently, FORS devices with optical fibres have replaced their mirror-based predecessors, rendering possible the realization of kilometre-long waveguides simply by wounding the fibre on carcasses with a diameter of several tens of centimetres [7].

Experimental findings show that the values of optical fibre refractive indices depend on temperature, and their temperature sensitivity can be as high as 3 × 10^−5^ K^−1^ [8,9]. Moreover, the effects of temperature on the fibre length and the wave propagation time through fibres also vary with the thermal coefficient of delay values, reaching 40 ps/(km × K) [10]. Thus, temperature gradients occurring in different parts of a fibre loop may produce errors in the rotation rate measurements. Furthermore, the existence of temperature differences elsewhere in the device might induce the potential risk of time-dependent temperature fluctuations, which increase the low-frequency noise in the device output [11].

Thermal fluctuations also affect FORS operation, causing the drift phenomenon known as the Shupe effect. For the above reasons, the sensor loop is prepared as a quadrupolar symmetrical winding, with the additional switch-on time for inner temperature gradient minimization. However, even if the conventional Shupe effect is successfully suppressed, the thermal-phase noise still may limit the FORS performance [12]. This type of noise is caused by spontaneous refractive index fluctuations and fibre-length fluctuations [13]. In fact, it has been shown that the thermal phase noise may even become the dominant noise source for fibre loops longer than 5.7 km [14]. Therefore, reducing the temperature and temperature gradients in FORSs is crucial to lowering the total noise floor and ensuring high accuracy.

Moreover, FORS devices are expected to work in various ambient conditions; standing in air and cooled by the natural convection, buried under the ground or even submerged in water, so their thermal models should be easily adaptable to the surrounding environment. Finally, temperature changes might be caused also by reasons related to software since the power dissipation depends on the implemented algorithm complexity [15].

In conclusion, it would be desirable to have a FORS device thermal model that could predict temperatures in the locations of interest for any power dissipation pattern and in various cooling conditions so as to assure its high sensitivity. This paper presents such a thermal model developed for the practical realization of the device measuring the rotation speed in three perpendicular directions. Compared to the original model presented in [16], most of its subcomponents have been modified and their element values fitted to new, larger series of temperature measurements. The following sections of this paper present the FORS device operating principles and its practical realization. Then, the main heat-flow paths in the device are determined, and its static thermal model is proposed. Next, the model element values are determined based on measurements. Finally, some important conclusions are drawn, providing indications for further research.

## 2. Device Operating Principle and Practical Realization

The FORS device, as mentioned in the introduction, uses the Sagnac effect to produce an interference fringe pattern by two light beams traveling through an optical fibre in opposite directions. The phase difference Δ*Φ_R_* between the beams is proportional to the rotation speed Ω and is equal to [1,17](1)$${∆\phi }_{R}=\frac{2\pi LD}{\lambda c}\Omega$$
where *L* is the fibre length, *D* is the coil diameter, *λ* is the wavelength and *c* is the speed of light in a vacuum.

Taking into account that the optical power of the fringe pattern is proportional to cos (Δ*Φ_R_*), an additional bias should be provided by the square-wave modulation having the period of 2Δ*τ_g_*, where Δ*τ_g_* is the group transit time around the fibre-optical loop. This is realized by placing at one end of the loop a phase modulator acting as a delay line and providing the bias required to adjust the operating point. Except for the fact that the FORS sensitivity increases with the difference in the length of the beam optical paths, several other concepts, both hardware and software, could be used in order to further improve its sensitivity. Regarding the hardware, the FORS minimal configuration assuring a full-reciprocal design might eliminate many negative effects related to the occurrence of parasitic interference fringes or the phase drift [18,19]. Considering the software, the algorithm, e.g., implemented in an FPGA, maintains the operating point close to the optimal one regardless of the rotation speed, exercises control over the phase modulator, filters out spikes that may appear in the power response due to the inaccuracy of the modulation wave period and generates and resets a digital ramp signal dependent on the phase difference. The details concerning the control algorithm can be found in [15].

A functional block diagram and photos of the FORS device investigated here are shown in Figure 1 and Figure 2. All the components are enclosed in a cubical metal housing having a volume of approximately one cubic foot (see Figure 2a). An Amplified Spontaneous Emission (ASE) C-M light source [20] provides light beams having wavelengths of 1.55 μm coupled to three Corning^®^ (Corning, NY, USA) SMF-28e+^®^ fibre-optic loops measuring the rotation speed in perpendicular directions. The loops, such as the one presented in Figure 2c, are located at the bottom, right and back walls. They are 6030 m, 5837 m and 6459 m long and wound around aluminium carcasses having diameters of 23.7 cm. Both the modulator and sensor modules are mounted inside the loops. The modulators contain Multifunctional Integrated Optical Chips, Optilab (Phoenix, AZ, USA) MIOC-1550-22-PG [21], which control the phase of the light waves. These chips are made on a lithium niobate substrate, such as the ones discussed in [22]. The sensor modules use InGaAs Avalanche Photo-Diodes (APDs) (see for details [23]) to measure the power of the interfering light waves.

The entire device is powered via a Power over Ethernet (PoE) 48 V DC input. The power module, pictured in Figure 2b, contains DC/DC converters adjusting the input voltages to the levels required by the other circuits. Based on the APD sensor signals, the control algorithm implemented in the FPGA circuit contained in the control module, also visible in Figure 2b, adjusts the light beam phase in the modulators. This is accomplished, as already mentioned, by filtering the input signals and generating all the necessary reset and ramping output signals. Ultimately, the control module computes the three components of the rotation speed vector.

## 3. Device Static Thermal Model

The FORS device usually operates continuously in stable ambient conditions; hence, only a static thermal model for this device was considered. Due to the complex geometry, it is virtually impossible to compute the temperature distribution inside the housing by FEM simulations. Thus, similarly as in the standard DELPHI-style models [24], a simplified model in the form of a thermal resistor network was adopted, where temperature values are computed only in selected nodes and element values are determined based on temperature measurements.

In developing a static thermal model of the FORS device, all main heat sources and heat-flow paths had to be identified. This was accomplished by performing a series of infrared measurements with the device housing open. Some examples of recorded thermograms for the power, modulator and sensor boards are presented in Figure 3. These measurements allowed the identification of the main heat sources in these circuits. Furthermore, the temperature value of the processor chip was read from its internal sensor.

Then, static thermal models consisting of resistor networks were created separately for each of the FORS device components and then connected together, forming a final model valid for the entire device. All these thermal model subcomponents, shown in Figure 4, have two common nodes: the ambient (marked by the ground node symbol) and *T_mid_*, referring to the interior of the device housing.

### 3.1. Power and Control Board Model

The first of the thermal models, presented in Figure 4a, corresponds jointly to the power and control boards, which are mounted next to each other on the front wall of the device. Most of the power *P_Pow_* dissipated in the power board flows to the outside wall, while the remaining heat flows to the interior of the device housing. Therefore, two thermal resistances were used to model these phenomena, *R_Pow-W_* and *R_Pow-Mid_*, respectively. However, the measurements show that due to the proximity of the control board, the direct heat transfer between the two boards is non-negligible. Therefore, an additional resistance *R_Ctrl-Pow_* was added into the model to represent the thermal coupling between these boards.

The part of the model corresponding to the control board is slightly more complex. The processor present in this board is cooled by a copper radiator screwed directly to the front wall, and, consequently, heat generated in this component flows almost uniquely through this thermal path. Thus, in order to properly model the heat flow in this module, two separate heat sources *P_Proc_* and *P_Ctrl_* were used, where the latter one reflects the power dissipated in this board elsewhere than in the processor. This direct heat conduction from the processor to the wall is modelled by the resistance *R_Proc-W_*, whereas the heat generated in the control board outside the processor, similar to the power board, flows either to the outside wall or to the interior of the housing, which is represented in the model by the resistances *R_Ctrl-W_* and *R_Crtl-Mid_*.

### 3.2. Light Source Model

The ASE light source is mounted on the left wall of the device. In its thermal model, shown in Figure 4b, the dissipated power *P_LS_* is represented by a current source, which is connected to the wall by the *R_LS-W_* thermal resistor. Additionally, the heat flow towards the device interior is modelled by the *R_LS-Mid_* resistance.

### 3.3. Horizontal Loop Model

The bottom loop sensing in the vertical Z direction is placed on the bottom wall of the device. The source P_loop,Z_ reflects the power dissipated both in the modulator and sensor modules. Since the measurements show that the loop temperature is almost uniform, it is represented in the thermal model shown in Figure 4c by a single node *T_LC_*_,*Z*_. Similar to the case of the light source, the heat from this loop can flow either to the bottom wall through the resistor *R_LC-W_* or towards the interior of the device housing via the resistor *R_LC-Mid_*. The convection resistance between the air inside the loop and its casing is modelled by the resistance *R_Int_*_,*Z*_.

### 3.4. Vertical Loop Models

The vertically mounted loops measuring rotation in the X and Y directions require more complex thermal models than the horizontal one due to the fact that there exist non-negligible thermal gradients between the loop top and bottom ends. These gradients are important from the point of view of FOG device performance because they translate to different temperature values in various parts of the optical fibre and may influence the total optical power output detected by the APD, which in turn affects the overall device accuracy. Consequently, as presented in Figure 4d,e, two nodes in the thermal model are used to represent the temperature of the loop, one corresponding to the temperature rise at the top of the loop and the other one to the temperature rise at the bottom. Taking into account that all three loops in the device have the same the geometry and the heat-flow path is split in the thermal models of vertical loops in two, the resistances *R_LC-Mid_* are doubled here to maintain the same resistance towards the housing interior as in the case of the horizontal loop.

### 3.5. Housing Model

The measurements revealed that the heat conduction through the device metal housing walls is an important factor determining the temperature distribution. Thus, the cube-shaped housing is modelled, as shown in Figure 4f–h, by twelve identical resistances *R_Case_* connecting six nodes representing the walls. Hence, each node has four interconnections to neighbouring walls.

### 3.6. Device Cooling Model

The FORS device housing has six almost equally sized walls through which heat is exchanged with the ambient air. Each wall is represented in the thermal model by a node, as pictured in Figure 4i–n, whose temperature rise equals the average temperature rise value at a given wall. Taking into account that during measurements the device was cooled by natural convection, all the wall ambient thermal resistances *R_W-Amb_* were assumed to have the same value, except for the resistance related to the bottom wall, which has a slightly higher value due to the fact that the device is placed on three very short legs and the heat exchange at this wall is hindered by restricted air movements. The change of cooling condition, e.g., when the device is buried under the ground, requires only the adjustment of the *R_W-Amb_* thermal resistance values.

## 4. Temperature Measurements and Simulations

This section of the paper presents the results of FORS device measurements and thermal simulations. First, the power dissipated in each component was determined and the initial values of resistors in the thermal model proposed in the previous section were determined based on the data on device geometry and material thermal properties. Then, the resistor values were fitted based on temperature measurement results.

### 4.1. Determination of Power Dissipation

The thermal model of the FORS device described in the previous section should correctly predict temperatures in various locations regardless of the current power dissipation pattern. Obviously, the power dissipation in various device modules cannot be arbitrarily changed, but in practice it may be varied to some extent. Therefore, all power dissipation and temperature measurements presented in Section 4 were carried out for four scenarios, S3, S2, S1 and S0, when three, two, one or none of the fibre-optic loops were active, respectively. When only one loop is active in scenario S1, the vertical loop at the right wall measuring the rotation in the X direction is turned on, while in scenario S2 with two loops active, the horizontal loop at the bottom wall measuring the rotation in the Z direction is additionally powered on.

These four scenarios differ in terms of power dissipation in the following manner: when a particular loop is deactivated, the sensor and modulator boards connected to this loop are switched off and do not consume any power. Additionally, turning off individual loops also changes the power dissipated by the power board. The total power dissipated inside each active loop *P_loop_*, being the sum of power values dissipated in the sensor and modulator boards, could be precisely measured by the FORS device, and it was equal to 2.45 W. On the other hand, the power dissipated in the power board *P_Pow_* could be precisely calculated for each of the four analysed scenarios knowing the energy efficiency of this board and the total power delivered from it to all other boards.

Although the total power dissipated in the control board can be measured, this value also includes the power of the light source located elsewhere but powered from the control board. Hence, the power dissipated in this source *P_LS_* had to be estimated based on the data provided by its manufacturer and subtracted from the total power dissipated in the control board. Additionally, as already mentioned in Section 3, there exist two separate heat transfer paths from this board, thus two power sources were used; *P_Proc_* representing the power dissipation in the main processor and *P_Ctrl_* reflecting the heat generated in the remaining parts of the board. Since it was not possible to measure directly the power dissipated in the processor itself, the total control board power value was divided between the two sources only during the model fitting. All the data concerning the power dissipated in the heat sources of the device thermal model are summarized in the Table 1.

### 4.2. Temperature Measurements

The temperature data were collected during measurements using a TC-08 data logger manufactured by Pico Technology (St Neots, UK) [25], recording data from eight K-type thermocouples. During the measurements, seven probes were inserted into the device, while the remaining one was recording the ambient temperature, which was then deducted from the other measurement data so as to later process only the temperature rise values.

The results of the temperature measurements recorded in 13 locations of interest for all the scenarios considered here are presented in Table 2 in the respective columns. The measurement locations correspond to the thermal model nodes. Taking into account that in the case of the FORS device it is crucial to determine the temperature differences of the fibre-optic loops, two measurement locations were chosen for both vertical loops. For the nodes that in reality represent the temperature of a larger area, measurements were taken in 2–3 different locations, and then the average temperature value was considered as the final measurement result. For example, for the device walls, their temperature values at the top and bottom were measured, and the average value was computed to minimize any error resulting from the existing spatial temperature gradient.

The measurements were time consuming because the moving of temperature probes to other locations required the opening of the device housing. Then, due to the high thermal capacity of the entire device, 6–8 h was required after the power-on until temperature values in all locations reached their steady state. On the other hand, turning the loops on and off did not require the removal of the device housing; therefore, only 1–2 h was needed to reach the steady state.

Looking at the data presented in Table 2, the highest temperature rise value, reaching 41.2 K, measured for the case with the highest power dissipation was recorded on the power board, which might indicate that in the future, better cooling should be considered for this module. The measured temperature rise value inside the FORS device housing *T_Mid_* was between 8.7 K and 11.8 K, the temperature rise value of the fibre loops reached 7.1 K and the temperature rise value of the light source was 15.4 K. The processor temperature rise, despite its relatively high power consumption, was fairly stable regardless of the scenario and did not exceed 11.7 K.

### 4.3. Thermal Simulations

The FORS device thermal model was implemented in the Analog Devices (Wilmington, MA, USA) LTspice^®^ environment. The model fitting was performed iteratively. Starting with the initial resistor values, thermal simulations helped to identify nodes where the errors were the highest. Next, the values of selected resistors were tuned in order to minimize these errors. Then, the entire procedure was repeated. Regardless of many conflicting objectives, owing to the deep understanding of thermal couplings between the device components, it was possible to achieve the desired simulation accuracy. The final values of all thermal resistors in the FORS device model are given in Table 3.

Looking again at Table 2 comparing the measured and simulated temperature rise values, it can be concluded that the proposed thermal model describes the device temperature very well. The average absolute error calculated for 52 simulated values is 0.49 K and the error was always below 1 K, except for one value equal to 1.62 K. This exception occurred for the front wall temperature in scenario S3 with all loops active. The explanation of this fact is straightforward, since the front wall is the only one where non-negligible spatial thermal gradients exist. Namely, the temperature is higher near the top, where the power board is mounted, and lower at the bottom. The temperature of this wall is represented by just one node; thus, the temperature value in this location is modelled less accurately, but it is not particularly important for the potential device performance degradation.

Additionally, the thermal resistor network model allows not only calculating node temperatures but also the heat flow between the model nodes. Therefore, after running the simulations, the heat-flow values between particular device parts were calculated, which allowed determining the main heat-flow paths in the device and the identification of thermal couplings among device components. For instance, it was observed that 72% of the heat generated in vertical loop X flows to the wall on which it is mounted, while only 28% flows to the interior of the housing. For the light source, this ratio was 55% to 45%, respectively. For the combined power dissipated in the power and control boards, the model showed that 69% of the heat flows to the device wall, whereas 31% heats up the device interior.

## 5. Conclusions

The static thermal model of the FORS device investigated in this paper allowed the calculation of temperature values in the selected locations of interest with satisfactory accuracy in a short time. The estimation of the model element values and the validation of the entire model was possible owing to the temperature measurements taken simultaneously in numerous locations inside the device housing and at its outer surfaces, which is the main novelty of the presented research. Taking into account that the device is expected to operate in various environmental conditions, e.g., buried underground or even submerged in water, the change of the surroundings can be easily taken into account by simply varying the values of the six thermal resistors modelling the heat exchange between a wall and the ambient air.

Looking at possible ways of further improving the accuracy of the thermal model, first of all it should be noted that the temperature distribution in the device walls is not uniform, especially in the front wall, where the power and control modules are located. The wall temperature values are represented in a model by just one resistor reflecting the average wall temperature values. On the other hand, with the convective air cooling, the values of these resistors depend quite strongly on the temperature differences between walls and the ambient air due to the change in the heat transfer coefficient value. Thus, introducing more nodes representing wall temperature values and modifying wall interfaces with modules attached to them could improve the model accuracy.

Furthermore, the air inside fibre-optic loops can move freely, and the exact positions of the modulator and sensor boards with respect to the device housing are never the same, since the loop casings might be differently rotated in each copy of the FORS device. Then, the boards can be placed in space one upon another or one next to each other. Consequently, the air temperature values vary inside loop casings, especially those vertically placed. Additionally, it is very difficult to measure air temperature values inside loops because the thermocouple locations are not precisely known, which is a source of measurement uncertainty.

Therefore, it is planned that in the next generation of FORS device prototypes developed in a new project starting this year, the thermocouples will be embedded in predetermined locations inside the loops. This will allow the investigation of the influence of loop casing rotation on temperature and the determination of the optimal board placement. Owing to these experiments, it will also be possible to capture tiny temperature differences inside the optical fibre casings and make the loop thermal models more detailed and accurate, e.g., by using separate power sources for the modulator and sensor boards and modelling the heat transfer between them in the loop interior.

## Figures and Tables

**Figure 1 sensors-25-03184-f001:**
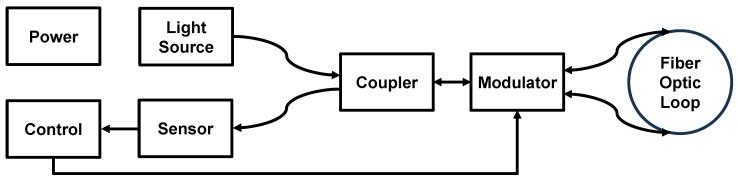
The FORS device functional block diagram.

**Figure 2 sensors-25-03184-f002:**
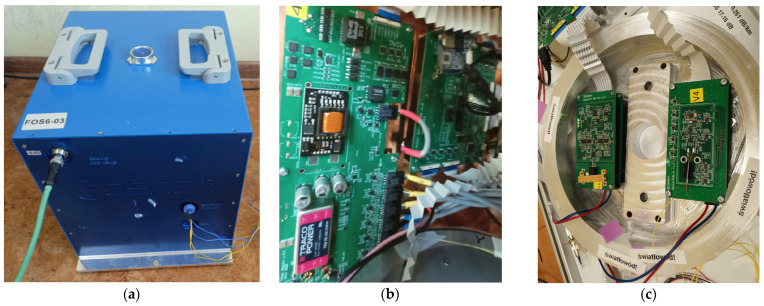
Photos of the FORS device: (**a**) general view of the housing; (**b**) front wall with power (left) and control modules (right); (**c**) fibre-optic loop with modulator (left) and sensor modules (right).

**Figure 3 sensors-25-03184-f003:**
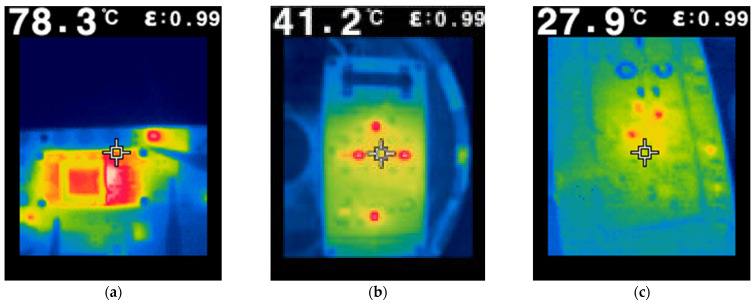
Infrared thermograms: (**a**) power module; (**b**) modulator circuit; (**c**) sensor board.

**Figure 4 sensors-25-03184-f004:**
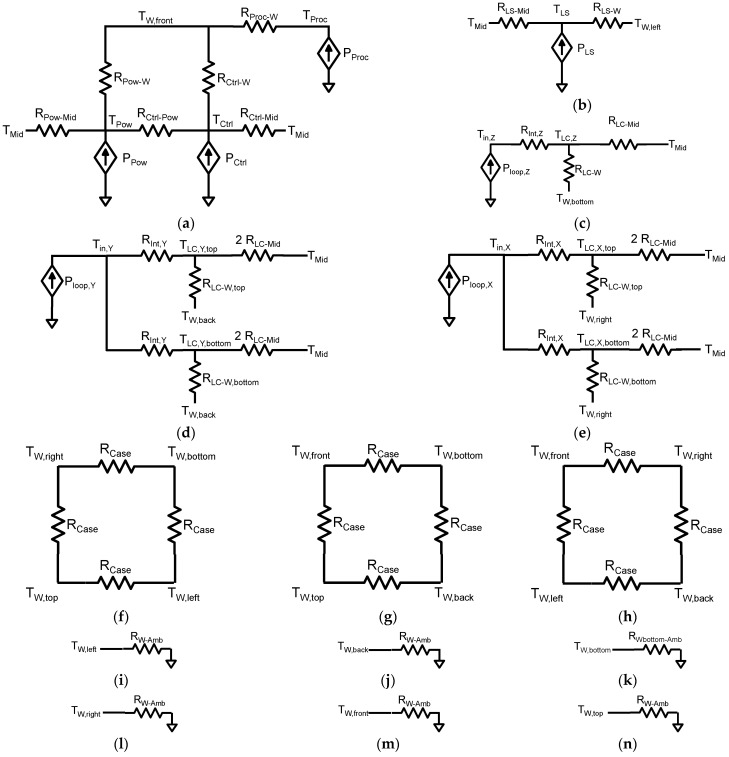
Thermal model subcomponents: (**a**) control and power modules at the front wall; (**b**) light source at the left wall; (**c**) fibre-optic loop at the bottom wall sensing in the Z direction; (**d**) fibre-optic loop at the back wall sensing in the Y direction; (**e**) fibre-optic loop at the right wall sensing in the X direction; (**f**) right wall cooling model; (**g**) front wall cooling model; (**h**) top wall cooling model; (**i**) wall model in the horizontal XY plane; (**j**) wall model in the vertical XZ plane; (**k**) wall model in the vertical YZ plane; (**l**) left wall cooling model; (**m**) back wall cooling model; (**n**) bottom wall cooling model.

**Table 1 sensors-25-03184-t001:** Heat source power dissipation data.

Heat Source	Scenario S3 (All Loops on) [W]	Scenario S2 (Loops X, Z on, Loop Y off) [W]	Scenario S1 (Loop X on, Loops Y, Z off) [W]	Scenario S0 (All Loops off) [W]
P_loop,X_	2.45	2.45	2.45	0.00
P_loop,Y_	2.45	0.00	0.00	0.00
P_loop,Z_	2.45	2.45	0.00	0.00
P_Pow_	3.00	2.58	2.16	1.74
P_LS_	2.00	2.00	2.00	2.00
P_Ctrl_	2.75	2.75	2.75	2.75
P_Proc_	5.50	5.50	5.50	5.50
Total	20.60	17.73	14.86	11.99

**Table 2 sensors-25-03184-t002:** Measured and simulated temperature rise data.

Temperature Rise	Scenario S3 [K]		Scenario S2 [K]		Scenario S1 [K]		Scenario S0 [K]	
	Meas	Sim	Meas	Sim	Meas	Sim	Meas	Sim
T_Mid_	11.8	11.9	10.4	10.5	9.5	9.1	8.7	7.8
T_Pow_	41.2	40.9	37.0	37.4	34.8	33.9	30.2	30.4
T_Proc_	11.7	11.0	10.3	10.7	9.9	10.3	9.3	9.9
T_LS_	15.4	14.6	14.3	13.5	13.0	12.3	11.9	11.2
T_LC,X,top_	5.6	5.2	3.9	4.7	4.0	4.3	2.8	2.3
T_LC,X,bottom_	7.1	6.5	5.6	5.9	4.6	5.3	3.3	2.7
T_LC,Y,top_	4.9	4.9	2.7	3.0	2.7	2.5	2.3	2.0
T_LC,Y,bottom_	6.9	6.3	4.5	3.7	3.7	3.0	3.3	2.5
T_LC,Z_	6.6	6.5	5.2	5.9	3.7	3.4	3.4	2.8
T_W,front_	6.3	4.7	5.2	4.4	4.2	4.0	3.4	3.6
T_W,back_	2.8	3.3	2.3	2.3	1.8	1.9	1.4	1.5
T_W,right_	3.2	3.6	2.8	3.2	1.9	2.8	1.3	1.7
T_W,left_	3.2	2.5	2.5	2.2	2.0	1.8	1.7	1.6

**Table 3 sensors-25-03184-t003:** Thermal model resistor values.

Thermal Submodel	Resistor	Value [K/W]
Outside case	*R_Case_*	2.5
	*R_W-Amb_*	0.9
	*R_Wbottom-Amb_*	1.6
Light source	*R_LS-Mid_*	3.0
	*R_LS-W_*	11.0
Power/control boards	*R_Pow-Mid_*	35.0
	*R_Pow-W_*	70.0
	*R_Ctrl-Mid_*	10.0
	*R_Ctrl-W_*	20.0
	*R_Ctrl-Pow_*	10.0
	*R_Proc-W_*	1.2
Fiber loops	*R_LC-Mid_*	8.0
	*R_LC-W_*	1.2
	*R_LC-W,top_*	0.7
	*R_LC-W,bottom_*	1.7
	*R_Int,X_*	3.0
	*R_Int,Y_*	4.5
	*R_Int,Z_*	2.0

## Data Availability

The data presented in this study supporting the conclusions of this article might be made available by the authors on request.

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
