# Peer review of "Static Thermal Model of a Fibre-Optic Rotational Seismometer"

_sensors, 2025, doi:10.3390/s25103184_

Round 1

Reviewer 1 Report

Comments and Suggestions for Authors

In this manuscript, the authors construct equivalent models of the components of a fiber-optic rotational seismometer in the form of a thermal resistor network, and obtain a thermal model of the seismometer in a stable state based on measured temperatures. The temperature variations simulated by the model are basically consistent with the experimental measurements. This model provides a reference for analyzing the internal heat flow process of a fiber-optic rotational seismometer. The perspective presented in the manuscript is interesting, and I believe this work is worth publishing. However, before acceptance, the following issues need to be addressed:

1. The expression in the first sentence of the introduction is inaccurate. The authors mention “measurements of rotation rate,” but fiber-optic seismometers include not only rotational seismometers, but also systems such as DAS and fiber-optic accelerometers, which are used to measure stress and translational motion. Please clearly define the scope of this manuscript.

2. Based on the context of the manuscript, it appears that the focus is specifically on fiber-optic rotational seismometers. Therefore, the authors are advised to consistently revise “fiber-optic seismometers” to “fiber-optic rotational seismometers” in the title, abstract, and throughout the text to ensure clarity and rigor.

3. The introduction lacks a discussion of recent developments in thermal effects for fiber-optic seismometers. For example, DOI: 10.1063/5.0135848 discusses the thermal phase noise, which is identified as a dominant factor in high-sensitivity fiber-optic seismometers.

4. Eq. (1) appears to be incorrect. The inclusion of n is unnecessary, as the Sagnac effect is independent of the refractive index of the medium, as explained in Ref. [1]. Although Ref. [14] is cited as the basis, the current manuscript introduces an extra n without justification. This should be corrected.

5. In Line 82, the sentence “They are 5837 m ÷ 6459 m long” is unclear. What does the symbol "÷" represent in this context?

6. Please check the temperature units used in Lines 251–257 and Table 2. Based on the units shown in Fig. 3, it seems that the temperature values are in degrees Celsius, not Kelvin.

7. While the manuscript establishes thermal submodels for individual components, these components exist within a common enclosed seismometer housing. Has the thermal coupling or heat exchange between components been considered in the model? The potential impacts should be discussed.

8. Lines 281–287 provide an explanation of the obtained model, which is a critical part of the analysis. However, the explanation lacks clarity. On what basis do the authors estimate the proportions of heat flow? How are these proportions derived from the data in Table 3? A more detailed explanation is necessary.

9. The term "DELPHI" appears abruptly in the abstract and may be unfamiliar to readers in the field of fiber-optic seismometers. Since the model is fundamentally based on a thermal resistor network, it is recommended that the term “DELPHI” be replaced with “thermal resistor network” to improve clarity. Accordingly, it is also suggested to revise the title to: “Static thermal resistor model of a fiber-optic rotational seismometer.”

Reviewer 2 Report

Comments and Suggestions for Authors

eport on “Static Thermal Model of a Fibre-Optic Seismometer.” In this contribution, the authors developed a thermal model of a fiber-optic seismometer device. The result can be interesting for scientists working in the optical fiber field. In my opinion, this paper can be considered for publication after revision. Please find below my comments and suggestions on how to improve the quality of the paper.

  1. The experimental set-up description is not strict enough. There is a lack of description of Athe SE source or modulator used in FOS.
  2. The authors should describe more about how temperature can affect the FSO performance, especially how it affects the optical fiber
  3. The authors state, “Experimental findings show that the values of optical fibre refractive indices depend 30 on temperature and their temperature sensitivity can be as high as 3 × 10-5 K-1”. But it is not clear what kind of type of fiber is used in the experiment. SMF-28 or maybe polarization maintaining.
  4. Moreover, the authors should explain whether there is any novelty in their approach to model the thermal distribution of a Fiber-Optical Seismometer, taking into account that they use commercial software
  5. For example, right now accurate temperature of optical fiber can be measured using a frequency domain  reflectometer 

Round 2

Reviewer 1 Report

Comments and Suggestions for Authors

The authors have adequately addressed the concerns. I recommend acceptance in its current form.